# Bioassay-Guided Isolation of Anti-Inflammatory Constituents of the Subaerial Parts of *Cyperus articulatus* (Cyperaceae)

**DOI:** 10.3390/molecules27185937

**Published:** 2022-09-13

**Authors:** Domenic Mittas, Monizi Mawunu, Giorgia Magliocca, Thea Lautenschläger, Stefan Schwaiger, Hermann Stuppner, Stefania Marzocco

**Affiliations:** 1Institute of Pharmacy/Pharmacognosy, Center for Molecular Biosciences Innsbruck, University of Innsbruck, Innrain 80/82, 6020 Innsbruck, Austria; 2University of Kimpa Vita, Province of Uíge, Rua Henrique Freitas No. 1, Bairro Popular, Uíge, Angola; 3Department of Pharmacy, University of Salerno, Via Giovanni Paolo II 132, I-84084 Fisciano, Italy; 4Department of Biology, Institute of Botany, Faculty of Science, Technische Universität Dresden, 01062 Dresden, Germany

**Keywords:** Piri Piri, stilbene oligomers, *trans*-scirpusin B, cyperusphenol B, inflammation, NO production

## Abstract

Based on data from a previous ethnobotanical study in northern Angola, phytochemical investigations into the methanolic rhizomes and roots extract of *Cyperus* *articulatus*, monitored by in vitro assays, resulted in the recovery of 12 sesquiterpenes, 3 stilbenes, 2 phenolic acids, 1 monoterpene, and 1 flavonoid. Among them, 14 compounds were isolated for the first time from this species. Their inhibitory potential against nitric oxide (NO) production, as well as inducible nitric oxide synthase (iNOS) and cyclooxygenase-2 (COX-2) expression, was evaluated in LPS-treated J774A.1 murine macrophages. Especially, both stilbene dimer *trans*-scirpusin B and trimer cyperusphenol B showed promising inhibitory activity against the production of the inflammatory mediator, NO, in a concentration-dependent manner (10–1 µM). The obtained data are the first results confirming the anti-inflammatory potential of *C.* *articulatus* and support its indigenous use as a traditional remedy against inflammation-related disorders.

## 1. Introduction

The sedge plant *Cyperus articulatus* (L.) (Cyperaceae), commonly known in the Amazonia region as Piri-Piri, is native to the subtropical areas of the world and spreads throughout African countries such as Cameroon, Nigeria, and Angola to the Amazonia region and is common in wet habitats [1]. The decoction of its horizontal rhizomes and roots is used to gain traditional remedies to treat a number of diseases such as epilepsy, migraine, cough, fever, gastrointestinal disorders, malaria, and tuberculosis [2,3]. The rhizome essential oils are acclaimed for their multiple beneficial effects as anticonvulsant [4,5], antimalarial [6], antimicrobial [7,8,9], antiproliferative [10], and antioxidant [11]. The chemical composition of the apolar extracts from rhizomes and the roots of *C. articulatus* has been exhaustively investigated by GC-MS, and this has revealed the identification of terpenoids. The biological activities described are attributed mostly to sesquiterpenes, which represent the major components present [12,13,14]. According to a previous ethnobotanical survey in the province of Uíge in northern Angola, 6 informants out of 62 recommended the underground plant material of *C. articulatus* for anti-inflammatory indications, such as stomach pains, backache, foot infection, and toothache [15]. Pharmacological investigations of its methanol extract in murine macrophages showed promising inhibitory effects against NO production (96.24–36.09% inhibition) and COX-2 expression (85.37–42.45% inhibition) for six different concentrations, between 100–1.25 µg mL^−1^ [16]. Reports on the inflammation-related use of the rhizomes and roots opened a new research focus for *C. articulatus*. The increased production of NO and/or increased expression of iNOS is regarded as a marker for various inflammatory disorders. Excessive levels of NO express an immune-activated state, in which iNOS is up-regulated in response to inflammatory stimuli—these mediators also trigger the synthesis of cyclooxygenase—and may lead to damage of the host tissue, impairment of the inflammatory response, and the progression of diseases such as arthritis, atherosclerosis, cancer, ulcerative colitis, and sepsis [17]. In addition, the inducible form, COX-2, can be considered as the main source of prostaglandin E_2_ (PGE_2_)-generation and as a key enzyme for regulating inflammation; therefore, it is the target for nonsteroidal anti-inflammatory drugs (NSAIDs), such as acetylsalicylic acid and indometacin [18]

The anti-inflammatory properties of sesquiterpenoid compounds upon LPS-induced NO production, tumor necrosis factor (TNF)-α, and interleukin (IL)-6 release in RAW 264.7 murine macrophages were reported at concentrations partially below 10 µM (IC_50_) [19,20]. Due to the promising pharmacological activity of the methanol extract from the rhizomes and the roots of *C. articulatus*, this study aimed to conduct a bioactivity-guided isolation of active fractions to identify the active principles. The anti-inflammatory activities were evaluated against the production of the mediator NO, as well as iNOS and COX-2 expression in LPS-stimulated J774A.1 murine macrophages. Previous ethnopharmacological studies of *C. articulatus* showed promising anti-inflammatory activities within its methanolic extract and have aroused interest in investigating its so far unknown active constituents, with the purpose of encouraging the plants’ traditional use for inflammation-related disorders in northern Angola [15,16].

## 2. Results

The anti-inflammatory effect of the methanolic rhizomes and roots extract of *C. articulatus* was confirmed in a previous ethnopharmacological report [16]. Due to the promising pharmacological activity of the crude extract, the isolation and identification of its active principles are described in the present study. In preliminary screening, the obtained PE, DEE, PRC, EtOAc, 1-BuOH, and H_2_O fractions were analyzed for their inhibition of NO_2_^−^ release in LPS-stimulated J774A.1 murine macrophages at non-antiproliferative concentrations of 10, 5, and 1 µg mL^−1^ (Figure 1). The DEE fraction exhibited the highest anti-inflammatory activity and was selected, along with the also active PE fraction, for further classical activity-guided isolation. The latter contains the essential oil of the rhizomes, which was acclaimed for multiple beneficial effects, as described above; however, anti-inflammatory reports on the essential oil from *C. articulatus* are missing in the literature. The antiproliferative results indicated a significant activity for the PE fraction at concentrations above 12.5 µg mL^−1^ [21] (Appendix A).

Separation of the highly active DEE fraction by *NP*-MPLC, with its HPLC-UV chromatogram shown in Figure 2B, yielded subfractions A–R. The HPLC-UV chromatograms of subfractions A–R are presented in the Appendix A. Their inhibitory effects on NO production and COX-2 and iNOS expression in LPS-stimulated macrophages were analyzed at lower concentrations (5, 1, and 0.1 µg mL^−1^) to evaluate their concentration-dependent activity (results shown in Table 1). The potential antiproliferative activities of the obtained subfractions were determined using the MTT assay (Appendix A). The criterion for including a subfraction in one of the subsequent studies was initially a requirement of at least 40% inhibition among one of the three pro-inflammatory parameters, as the positive controls used yielded an inhibition of approximately 44% at a concentration of 1 µM, respectively. Considering the available amount and purity of the samples, subfractions E, F, H, M, N, and P were finally selected for detailed phytochemical and pharmacological investigations. Subfraction E showed promising inhibitory effects for NO production (31.33% at 5 µg mL^−1^), iNOS expression (33.67% at 1 µg mL^−1^) and COX-2 expression (43.00% at 5 µg mL^−1^). Subfraction F was of interest due to an antiproliferative effect of 15.33% on the J774A.1 murine macrophage at a concentration of 5 µg mL^−1^. Subfractions H and N revealed 32.33% and 57.67% inhibition for NO production in the macrophages stimulated by LPS; subfraction M exhibited a 35.0% inhibition during the iNOS expression inhibition assay, and subfraction P suppressed COX-2 expression by 40.33% at respective concentrations of 5 µg mL^−1^ vs. LPS alone. The major constituents of the selected subfractions, and the second most active PE fraction, were isolated as described in 4.4. The HPLC-DAD and GC analysis of the PE fraction displayed a rather complex composition with similar molecular ion peaks and an identical, expected polarity for the main compounds **1**–**6** at *m/z* 219.1–223.0 in the positive mode according to HPLC-ESI-MS. Thus, the compounds of the PE fraction were isolated without any intermediate pharmacological testing carried out on the subfractions, as described in Section 4.4.1.

The phytochemical investigation of the PE fraction, with its HPLC-UV chromatogram shown in Figure 2A, resulted in the identification of 11 sesquiterpenes, copa-3-en-2α-ol (**1**) [22], caryophyllene oxide (**2**) [23,24], humulene epoxide-II (**3**) [25,26], mustakone (**4**) [5,27], kobusone (**5**) [28,29], cyperotundone (**6**) [30], humulene dioxide (**7** and **8**) [31,32], (-)-guaia-1(10),11-dien-9-one (**9**) [33], muurolane-2β,9β-diol-3-ene (**10**) [34], and corymbolone (**11**) [35,36]. The PE fraction and its volatile compounds were additionally analyzed by gas chromatography, as explained in Section 4.2 (Appendix A).

The bioactivity-guided isolation of the most active DEE fraction, with its HPLC-UV chromatogram shown in Figure 2B, afforded two phenolic acids: *p*-hydroxybenzoic acid (**12**) [37] and *trans*-*p*-hydroxycinnamic acid (**13**) [38,39], one racemic flavonoid: 2*R*/2*S* dihydroluteolin (**14**) [40], two terpenoids: 4*R*/4*S*-4-hydroxy-1,10-seco-muurol-5-ene-1,10-dione (**15**) [41] and *trans*-sobrerol (**18**) [42], and three stilbenes: piceatannol (**16**) [43], *trans*-scirpusin B (**17**) [44,45], and cyperusphenol B (**19**) [46]. The NMR data for the compounds isolated are reported in the Appendix A. They were identified as known compounds by evaluation of their spectroscopic data in comparison with values found in the literature. Their structures and relative configurations are shown in Figure 3. Inhibition assays of the production of the mediator NO, as well as of iNOS and COX-2 expression in LPS-stimulated J774A.1 murine macrophages, were performed at different concentrations (10, 5, 1 µM) to evaluate the anti-inflammatory capacity of the eight identified constituents of the DEE fraction. As a positive control, 1 µM of L-*N*-nitro arginine methyl ester (L-NAME) and indometacin were used (results shown in Table 2).

## 3. Discussion

The COX-2 activity of fraction E, obtained at a concentration of 5 µg mL^−1^, was lost during the isolation processing of compounds **12**–**14**. In contrast to data from the literature, compound **12** showed a strong in vitro inhibition of NO production in the used assay at concentrations of 10 and 5 µM. The inhibitory potential of compound **13** was influenced by high antiproliferative effects that were revealed in the MTT assay [48]. Compound **14** suppressed iNOS expression at concentrations of 10 and 5 µM by 40.32% and 32.25%, respectively [49]. The antiproliferative properties of fraction F were not detectable for its isolated major component, sesquiterpene **15**. The in vitro anti-inflammatory activity of 4*R*/4*S*-4-hydroxy-1,10-seco-muurol-5-ene-1,10-dione is reported here for the first time, indicating moderate activities against NO production (40.27%) and iNOS expression (35.48%) at the highest test concentration of 10 µM. Although *trans*-sobrerol (**18**) was isolated from fraction N and can be found in the subaerial parts of other *Cyperus* species, such as *Cyperus longus* (L.) [50] or *Cyperus rotundus* L. [51], the use of extraction solvents may cause the formation of compound **18** from α-pinene oxide [52]. This may be a reason for the loss of activity during the isolation process, which occurred for subfraction N, and the observed cytotoxicity of its isolated compound **18** up to the lowest test concentration of 1 µM (Appendix A). In fraction M, the isolated stilbene dimer **17** (*trans*-scirpusin B, a dimer of piceatannol) was the dominant constituent, and it strongly attenuated the NO production in LPS-stimulated murine macrophages at all tested concentrations (46.50–31.65% inhibition vs. LPS alone). Additionally, COX-2 expression was reduced by 36.03% at a concentration of 10 µM. Apart from compound **17**, stilbene trimer **19** (cyperusphenol B, a trimer of piceatannol) was obtained as a highly active constituent from subfraction P. Treatment with compound **19** inhibited the NO production to a similar extent (as the tested stilbene dimer) at all tested concentrations (49.63–30.33% inhibition vs. LPS alone). The discrepancy in activity from the stilbene oligomers to the respective positive control was minimized at a test concentration of 1 µM. According to HPLC-DAD analysis, at 205 nm, stilbene oligomers **17** and **19** represent major components of the DEE fraction (10.2% and 8.4% of total integral, respectively) and are most likely the key contributors to the determined in vitro anti-inflammatory activity of the DEE fraction. Stilbene monomer **16**, isolated from subfraction H, revealed lower inhibitory effects on all tested pro-inflammatory parameters compared to stilbene dimer **17** and trimer **19**. Piceatannol (**16**) can be considered a well-known anti-inflammatory agent, showing a stronger in vitro inhibition of NO and PGE_2_ production (as well as NF-*κ*B activation in LPS-stimulated macrophages) than resveratrol [53]. Anti-inflammatory reports for compounds **17** and **19** are missing in the literature and are less abundant for other stilbene oligomers [54]. α-Viniferin, a trimer of resveratrol, has been reported to show inhibitory effects on NO and COX-2 production in LPS-activated RAW 264.7 cells (IC_50_ < 10 µM) [55]. A previous report assessed the effect of some stilbene oligomers to inhibit NO production in LPS-activated murine microglia BV-2 cells, indicating the partially higher inhibition potential of stilbene dimers and trimers over resveratrol [56]. As it appears from the results of the performed assays, the pharmacological activities of the DEE fraction and its compounds isolated are mostly mediated by the identified stilbene oligomers.

Compounds isolated from the DEE fraction exhibited the highest potency for inhibiting NO production. Thus, inhibition assays for NO production in LPS-stimulated J774A.1 murine macrophages were only carried out at three different concentrations (20, 10, and 5 µM) to evaluate the anti-inflammatory activities of the 11 identified sesquiterpenes of the second most active PE fraction. Unfortunately, the sesquiterpenes were almost completely inactive and showed superior antiproliferative effects at the tested concentrations (Appendix A). As compounds **7** and **8** are isomers of humulene dioxide, only compound **7**, available in a larger amount, was included in the assay and was identified as a very slight NO production inhibitor (31.50% at 20 µM vs. LPS alone). However, the anti-inflammatory effects of various essential oils, with a predominance of sesquiterpenes, have been reported in the literature. On the one hand, caryophyllene oxide (29.5%) (**2**), humulene epoxide II (4.2%) (**3**), and mustakone (1.2%) (**4**) represent the main constituents of the essential oil obtained from the leaves of *Allophylus edulis* Radlk., acclaimed for its multiple anti-inflammatory capacities in mouse paw oedema, cold allodynia, and hyperalgesia [57]. The essential oil of Algerian *Cyperus rotundus* L. is rich in sesquiterpenes, such as humulene epoxide II (21.3%) (**3**) and caryophyllene oxide (13.3%) (**2**), and reduces inflammation by inhibiting 5-lipoxygenase (5-LO) and leukotriene A4 hydrolase (LTA4H) [58]. On the other hand, NO production in BV-2 microglial cells was suppressed only slightly by kobusone (IC_50_ > 85 µM) (**5**) [59], and no inhibition effect on inflammatory cytokine (IL-6) release in murine macrophages could be noted [20].

All in all, the inhibitory activity of the PE fraction of the methanolic extract of *C. articulatus* on NO release in LPS-stimulated J774A.1 murine macrophages got lost during the fractionation process. The overall anti-inflammatory activity of the PE fraction could result from the potential additive or synergistic effects of sesquiterpenes or the effects of non-identified constituents. The beneficial interactions between multiple agents might be a pure summation effect, causing diminished biological activity in the isolated sesquiterpenes on their own, as observed in the NO production inhibition assay [60]. The data presented identified *trans*-scirpusin B (**17**) and cyperusphenol B (**19**) as the most active constituents in the methanol extract from the rhizomes and roots of *C. articulatus*. Interestingly, both compounds responsible for the observed biological activity were identified by HPLC-DAD and HPLC-ESI-MS and also in the PRC fraction of the methanolic extract. In comparison to the PE and DEE fractions, its inhibition of NO production was slightly lower in the preliminary screening model. According to HPLC-DAD analysis at 205 nm, it is likely that the dominant lipophilic area at the end of its chromatogram attenuated the activity of this fraction. Moreover, the HPLC-UV chromatogram of the PRC fraction was almost identical to that of the DEE fraction of the methanolic extract of the rhizomes and roots of *Scirpoides holoschoenus* L., except for the lipophilic region at the end (Appendix A). In comparison to the published data [47], the HPLC-DAD and HPLC-ESI-MS analysis indicated the presence of two stilbene dimers: scirpusin A (21) and cassigarol E (26), five stilbene trimers: cyperusphenol D (20), passiflorinol A (22), cyperusphenol A (24), mesocyperusphenol A (25), and passiflorinol B (23), and one ferulic acid derivative: smiglaside C (27) in the PRC fraction of the methanolic extract of *C. articulatus*, in addition to 2*R*/2*S* dihydroluteolin (**14**), piceatannol (**16**), *trans*-scirpusin B (**17**), and cyperusphenol B (**19**), which were already isolated and tested. Finally, the PRC fraction, containing many potentially active stilbene oligomers, might be interesting for further investigations as well.

## 4. Materials and Methods

### 4.1. Solvent and Reagents

VWR International (Darmstadt, Germany) supplied the reagents and solvents of an analytical grade. Solvents used for HPLC analysis were purchased from Merck (Darmstadt, Germany). A Sartorius Arium 611 UV water purification system (Sartorius AG, Göttingen, Germany) provided ultrapure water for the HPLC analysis. NMR deuterated solvents: acetone-*d_6_*, chloroform (chloroform-*d*), and methanol (MeOH-*d*_4_), were obtained from Euriso-top SAS (Saint-Aubin Cedex, France).

### 4.2. General Experimental Methods

Fractionation and isolation steps were carried out using both column chromatography systems, Sephadex LH-20 (Pharmacia Biotech, Uppsala, Sweden) and silica gel 60 (Merck KGaA, Darmstadt, Germany), as well as a Reveleris X2 MPLC system (Büchi, Flawil, Switzerland) with commercial prepacked columns. TLC was performed on precoated silica gel 60 F_254_ plates (Merck, Darmstadt, Germany, 1 mm thickness). Chromatographic experiments were performed using a HP1100 system (Agilent, Waldbronn, Germany), equipped with an auto-sampler, a DAD, and column thermostat. For HPLC-ESI-MS purposes, the system was hyphenated to an Esquire 3000 plus ion trap (Bruker Daltonics, Bremen, Germany) utilizing electrospray ionization (ESI). LC parameters for the petroleum ether fraction of the methanolic extract: stationary phase: YMC-Pack Pro-C_18_ 3 µm, 4.6 × 150 mm; mobile Phase: A = H_2_O + 0.9% formic acid (FA)+ 0.1% acetic acid (AA) and B = acetonitrile (ACN) + 0.9% FA + 0.1% AA for LC-MS, A = H_2_O + 0.02% trifluoroacetic acid (TFA) and B = ACN for LC-DAD; gradient: 0–25 min: 62.5% B, 40 min: 98% B, 55 min: stop; temperature: 15 °C; flow: 1 mL min^−1^; injection volume: 2 µL; sample concentration: 2 mg mL^−1^ in MeOH. LC parameters for the diethyl ether fraction of the methanolic extract: stationary phase: Phenomenex Synergi Polar RP-C_18_ 4 µm, 4.6 × 150 mm; mobile Phase: A = H_2_O + 0.9% FA + 0.1% AA for LC-MS, A = H_2_O + 0.02% TFA for LC-DAD, B = ACN; gradient: 0 min: 20% B, 25 min: 40% B, 30 min: 98% B, 35 min: stop; temperature: 40 °C; flow: 1 mL min^−1^; injection volume: 2 µL; sample concentration: 2 mg mL^−1^ in MeOH. ESI-MS parameters: 1:5 split from HPLC, dry temperature: 320 °C; dry gas: 12.0 L min^−1^; nebulizer: 25 psi; full scan mode: *m/z* 50–1200; ion polarity: alternating mode; capillary voltage: 4.5 kV.

The petroleum ether fraction of the methanolic extract was additionally analyzed using a Perkin Elmer gas chromatograph with FID (Waltham, MA, USA). GC parameters: capillary column: Permabond SE-54-DF, 25 m × 0.32 mm × 0.25 µm; temperature program: 130 °C (1 min), 130–180 °C (4 °C/min), 180 °C (5 min), 180–250 °C (20 °C/min), 260 °C (10 min), 36.5 min: stop; injection volume: 2 µL; split ratio: 5:1; split flow rate: 1 mL min^−1^; carrier gas: He; sample concentration: 1 mg mL^−1^ in hexane, injector temperature: 260 °C, detector temperature: 280 °C.

Semi-preparative HPLC experiments, as final isolation steps, were taken on a 1260 Infinity II (Agilent, Waldbronn, Germany) and a Dionex UltiMate 3000 (Thermo Fisher scientific Inc., Waltham, MA, USA), both equipped with an auto-sampler, a DAD, a column thermostat, and a fraction collector.

One- and two-dimensional NMR experiments were recorded on a Bruker Avance II 600/Bruker Ultrashield 400 Plus spectrometer (Bruker BioSpin, Rheinstetten, Germany), operating at 600.19 MHz/400.19 MHz (^1^H) and 150.91 MHz/100.62 MHz (^13^C) at 300 K (chemical shifts *δ* in ppm, coupling constants *J* in Hz). ECD spectra were recorded using a J-1500 circular dichroism spectrophotometer (JASCO, Tokio, Japan), using quartz cuvettes (d = 1 cm) and the optical rotations were measured using a JASCO P-2000 polarimeter.

### 4.3. Plant Material

The fresh rhizomes and roots of *Cyperus articulatus* were purchased from the village Kilomosso, in the province of Uíge, in northern Angola in February 2018 (S 7°38′31.0″, E 15°00′21.9″). The plant material was dried immediately after collection in a drying cabinet at 40 °C (HTD 100 Bench Top, LinTek) and stored in bags until extraction. Voucher specimens are deposited at the Herbarium Dresdense (DR), Institute of Botany, TU Dresden (voucher no. DR050752, DR051629).

### 4.4. Extraction and Isolation

The dried and milled rhizomes and roots of *C. articulatus* (239 g) were extracted with 125 mL methanol in the ultrasonic bath for 10 min. Subsequently, the filtrate was evaporated to dryness using a vacuum rotavapor with a waterbath at 40 °C. To assure exhaustive extraction, the procedure was repeated 10 times until colorlessness of the organic solvent to yield a total of 47.5 g crude methanolic extract. An amount of 45 g extract was suspended in 1 L HPLC water and was successively extracted four to six times with 500 mL portions, each of petroleum ether (PE), diethyl ether (DEE), ethyl acetate (EtOAc), and water saturated 1-butanol (1-BuOH) until exhaustion (yield after solvent removal: PE: 8.78 g, DEE: 2.20 g, EtOAc: 0.82 g, 1-BuOH: 3.45 g, H_2_O: 25.87 g). During the extraction with diethyl ether, an off-white precipitate was formed, which was separated and treated as additional fraction (PRC: 4.24 g).

#### 4.4.1. Isolation of Compounds **1**–**11** from the PE Fraction of the Methanolic Extract of *C. articulatus*

An amount of 8.2 g of the PE fraction was separated by silica gel column chromatography (CC, 135 g of silica gel 60, granulation: 40–63 µm) using varying mixtures of PE and acetone, starting from 100% PE as eluent, with increasing acetone amounts in steps of 5%. The eluate was collected in portions of 20 mL in test tubes and analyzed by TLC and subsequently combined into fractions 1–17. The separation of fraction 2 (2.1 g) by size exclusion chromatography (CC, Sephadex LH-20, Ø = 30 mm, l = 1000 mm), with isocratic elution using dichloromethane (DCM) and acetone (85:15 (*v*/*v*)), generated the subfractions 2A–2V. Subfractions 2E–2I (915 mg) and 2M–2O (274 mg) were fractionated on *NP*-MPLC, leading to subfractions 2E.A–2E.W and 2M.A–2M.N, using the following parameters: Reveleris NP column 40 g, 40 µm; mobile phase: A = hexane (Hex), B = methyl *tert*-butyl ether (MTBE) (9:1); gradient: 0–5 min: 100% A, 15–40 min: 90% A + 10% B, 45–50 min: 100% B, 50.1 min: stop; flow: 15 mL min^−1^; liquid sample injection: sample dissolved in Hex; detection: ELSD, UV (205, 220, and 320 nm); collection mode: collect all; tube volume: 5 mL (peak), 20 mL (non-peak). Fraction 2E.F (55 mg) was further processed by semi-preparative HPLC column chromatography (stationary phase: Merck LiChrospher RP-C_18_ 5 µm, 125 × 4 mm; mobile phase: ACN/H_2_O 70:30 (*v*/*v*), isocratic; temperature: 15 °C) to yield 1.9 mg of pure **3** and 8.5 mg of pure **2**. The same semi-preparative HPLC analysis afforded 37 mg of pure **1** for combined fractions 2M.E–2M.F (116 mg) and fractions 2E.G–2E.H (61 mg) provided 8.5 mg of pure **9**. Fractions 2E.M and 2E.N were combined (226 mg) and subjected to semi-preparative HPLC column chromatography (stationary phase: Phenomenex Synergi Polar RP-C_18_ 4 µm, 250 × 10 mm; mobile phase: ACN/H_2_O 45:55 (*v*/*v*), isocratic; temperature: 15 °C) to yield 10 mg of compound **6** and 51 mg of compound **4**. The use of semi-preparative HPLC column chromatography (stationary phase: Phenomenex Luna PFP(2) RP-C_18_ 5 µm, 250 × 10 mm; mobile phase: ACN/H_2_O 40:60 (*v*/*v*), isocratic; temperature: 15 °C) of fraction 2E.U (100 mg) yielded compound **5** (7.2 mg) and subfraction 2E.U.A (27 mg), which was separated by prep-TLC (glass plate 20 × 20 cm; stationary phase: PLC silica gel 60 F_254_ 1 mm; mobile phase: MeOH (pre-washing), DCM/Hex/acetone (6:3:0.5, all (*v*/*v*)), DCM (dissolving the compounds from the silica gel) to receive 8.4 mg of pure **7** (R_f_ 0.42) and 4.1 mg of pure **8** (R_f_ 0.50). Fractions 7 and 8 were combined (588 mg) and loaded onto a *NP*-MPLC system, as described above, however, using pure DCM as solvent. Among fractions 7A–7T, subfractions 7D–7E (65 mg) were finally purified by means of semi-preparative HPLC (CC, stationary phase: Phenomenex Synergi Polar RP-C_18_ 4 µm, 250 × 10 mm; mobile phase: ACN/H_2_O 50:50 (*v*/*v*), isocratic; temperature: 15 °C) to isolate 17.6 mg of pure **11**. Utilizing the same semi-preparative HPLC conditions and adjusting the solvents to a ratio of 35:65 (*v*/*v*), 11 mg of pure compound **10** was obtained from fraction 15 (200 mg). The purity of the isolated compounds was evaluated using HPLC-DAD analysis at 205 nm and exceeded 90% in all cases.

#### 4.4.2. Isolation of Compounds **12**–**19** from the DEE Fraction of the Methanolic Extract of *C. articulatus*

An amount of 2.1 g of DEE extract was separated by *NP*-MPLC, yielding fractions A–R (stationary phase: Reveleris NP column 40 g, 40 µm; mobile phase: A = DCM, B = acetone, C = MeOH; gradient: 0–5 min: 100% A, 60 min: 30% A + 70% B, 70–75 min: 100% B, 75.1 min: 100% C, 85 min: stop; flow: 15 mL min^−1^; solid sample injection; detection: ELSD; collection mode: collect all; tube volume: 5 mL (peak), 20 mL (non-peak)). Fraction E (100 mg) was chromatographed on Sephadex LH-20 (CC, Ø = 20 mm, l = 900 mm, elution with methanol) to yield 1.2 mg of compound **12**, 1.6 mg of compound **13**, and 1.7 mg of compound **14**. Fractions H (160 mg), M (68 mg) and P (80 mg) were purified analogously to receive 58 mg of pure **16**, 21 mg of pure **17,** and 32 mg of pure **19**. Fraction F (130 mg) and 38 mg of fraction N were further processed by semi-preparative HPLC (CC, stationary phase: Phenomenex Synergi Polar RP-C_18_ 4 µm, 250 × 10 mm; mobile phase: ACN/H_2_O 25:75 (*v*/*v*), isocratic; temperature: 15 °C) to afford 10 mg of pure **15** and 5.4 mg of compound **18**. The purity of the isolated compounds was evaluated using HPLC-DAD analysis at 205 nm and exceeded 90% in all cases.

### 4.5. Cell Culture and Pharmacological Assays

#### 4.5.1. Anti-Proliferative Assay

American Type Culture Collection (ATCC, Manassas, VA, USA) supplied the J774A.1 murine monocyte macrophage cell line used for assays. Cells were grown adherent to Petri dishes with Dulbecco’s modified Eagle’s medium (DMEM), which was supplemented with 10% fetal calf serum (FCS), 25 mM HEPES, 2 mM glutamine, 100 u/mL penicillin and 100 mg mL^−1^ streptomycin at a humidified air with 5% CO_2_ and 37 °C in an incubator to enable the cells to attach and reach the exponential phase of growth. Macrophages J774A.1 were plated into 96-well plates (5 × 10^4^/well) and adhered. The spent medium was removed and replaced with fresh DMEM alone or with serial dilutions of the fractions (100–12.5 µg mL^−1^), subfractions (5–0.1 µg mL^−1^), and isolated compounds **1**–**19** (20–1 μM) for 24 h. Relative cell viability was determined by 3-(4,5-dimethyltiazol-2yl)-2,5-phenyl-2H-tetrazolium bromide (MTT) assay [61]. Briefly, 25 μL of MTT at a concentration of 5 mg mL^−1^ was added, and the cells were incubated for additional 3 h. Once the cells lysed, the dark blue crystals were solubilized with 100 µL of a solution, consisting of 50% (*v*/*v*) *N*,*N*-dimethylformamide, 20% (mL L^−1^) sodium dodecyl sulfate (SDS), which was adjusted to a pH of 4.5. The amount of MTT reduction was measured by detecting the optical density (OD) in a microplate spectrophotometer (Titertek Multiskan MCC/340; Dasit, Cornaredo, Milan, Italy) at a wavelength of 550 nm and a reference wavelength of 620 nm.

Percentage J774A.1 macrophage viability in response to treatment was calculated using the following formula: % cellular inhibition = 100 − (OD treated/OD control) × 100.

#### 4.5.2. J774A.1 Murine Macrophage Cell Line

Macrophages J774A.1 were plated into 96-well plates (5 × 10^4^/well) and adhered. The spent medium was removed and replaced with fresh DMEM alone or with serial dilutions of the extracts (10–1 µg mL^−1^) for 1 h, and ultimately, the cells were co-exposed to a final concentration of LPS (1 µg mL^−1^) for an additional 24 h to detect the production of NO. The same protocol was used to assess the production of NO, and the expression of iNOS and COX-2 after treatment with fractions A–R and the isolated compounds **1**–**19** based on three concentrations, 5–0.1 µg mL^−1^ or 20–1 µM, respectively.

#### 4.5.3. *Measurement of NO Release*

After 24 h of LPS stimulation, NO levels were measured by Griess reaction in the culture medium of J774A.1 macrophages as nitrite (NO_2_^−^), the index of NO released by cells [62]. Briefly, the macrophages were exposed to samples and LPS; subsequently, 100 µL of cell culture medium were mixed with 100 µL of Griess reagent (equal volumes of 1% (*w*:*v*), sulphanilamide in 5% (*v*/*v*), phosphoric acid and 0.1% (*w*:*v*), and *N*-(1-napthyl)ethylenediamine dihydrochloride in water) and incubated at room temperature for 10 min. The absorbance was detected in a microplate reader (Titertek Multiskan MCC/340) at a wavelength of 550 nm. The amount of NO_2_^−^ measured in the samples is expressed as μM concentration, which was calculated via a sodium NO_2_^−^ standard curve.

#### 4.5.4. *Measurement of iNOS and COX-2 Expression*

After the J774A.1 macrophages were exposed to samples and LPS, they were processed as follows: collected, washed with phosphate-buffered saline (PBS), and first incubated in fixing solution (containing PBS, 2% FBS and 4% formaldehyde) for 20 min at 4 °C and then incubated in fix perm solution (containing PBS, 2% FBS and 4% formaldehyde and 0.1% Triton X) for another 30 min at 4 °C. Anti-iNOS/Anti-COX-2 (BD Laboratories, Milan, Italy) were added for 30 min. The secondary antibody was added in fix perm solution and finally, the cells were evaluated using a fluorescence-activated cell sorting (FACSscan; Becton Dickinson, NJ, Franklin Lakes, USA) and elaborated using Cell Quest software [63].

#### 4.5.5. *Data Analysis*

The data are expressed as mean values in % inhibition ± standard error of the mean (s.e.m), referred to as the LPS stimulated control group. At least three independent experiments, at least in triplicate, were performed for each sample and concentration.

## 5. Conclusions

The bioassay-guided isolation of the PE and DEE fractions of the methanolic rhizomes and roots extract of *C. articulatus* afforded the recovery of 12 sesquiterpenens (**1**–**11**, **15**), 2 phenolic acids (**12**, **13**), 1 flavonoid (**14**), 3 stilbenes (**16**, **17**, **19**), and 1 monoterpene (**18**). Among these constituents, compounds **1**, **5**, **7**–**10**, and **12**–**19** were isolated for the first time from this species. In summary, the stilbene dimer *trans*-scirpusin B (**17**) and trimer cyperusphenol B (**19**) showed promising inhibition over the production of the inflammatory mediator, NO, in LPS-stimulated J774A.1 murine macrophages and can be assumed to be the major active principle in the methanolic rhizomes and roots extract of *C. articulatus*, according to the in vitro assays performed. The moderate inhibitory activity of compounds **12**, **13**, **15,** and **16** against NO release, compounds **14**, **15,** and **16** against iNOS expression, and compounds **16** and **17** against COX-2 expression contribute to the anti-inflammatory potential of the extract. The data obtained through bioactivity-guided isolation, monitored by in vitro assays, are the first results confirming the anti-inflammatory potential of *C. articulatus* [16]. Further bio-guided studies on the PRC fraction, containing at least seven stilbene oligomers, might be needed to identify more of the active principles in *C. articulatus*. The mode of action and the in vivo activity of the active constituents: *trans*-scirpusin B (**17**) and cyperusphenol B (**19**), should be investigated to prove their suitability as anti-inflammatory candidates. In conclusion, the phytochemical and ethnopharmacological investigations of the rhizomes and roots extracts of *C. articulatus* support its use as a traditional remedy to treat inflammation-related disorders in northern Angola and other countries.

## Figures and Tables

**Figure 1 molecules-27-05937-f001:**
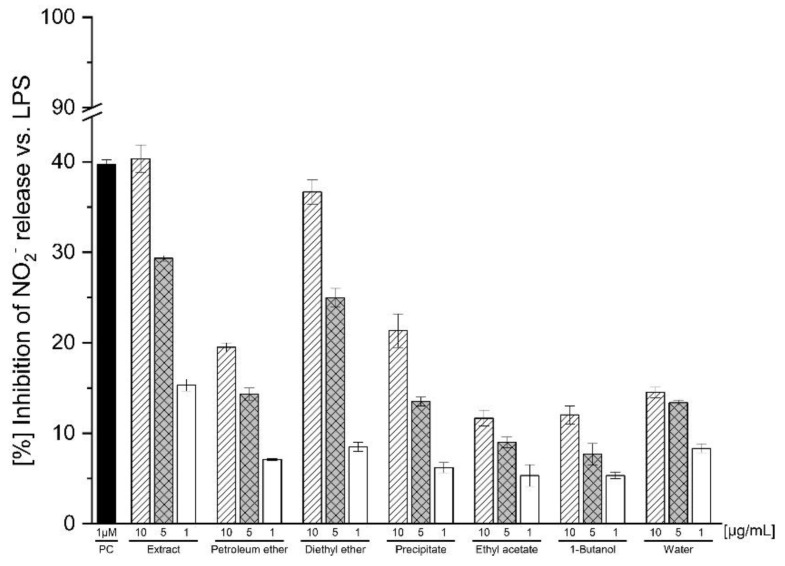
Inhibitory effects of the methanolic extract (Extract) and its subfractions (Petroleum ether, Diethyl ether, Precipitate, Ethyl acetate, 1-Butanol, and Water) obtained from the methanolic rhizomes and roots extract of *C. articulatus* on LPS-stimulated J774A.1 murine macrophages, which were incubated with subfractions at concentrations of 10, 5, and 1 µg mL^−1^, followed by measurement of the NO production vs. LPS ± SEM (*n* = 3). L-NAME was used as positive control (PC) at a concentration of 1 µM.

**Figure 2 molecules-27-05937-f002:**
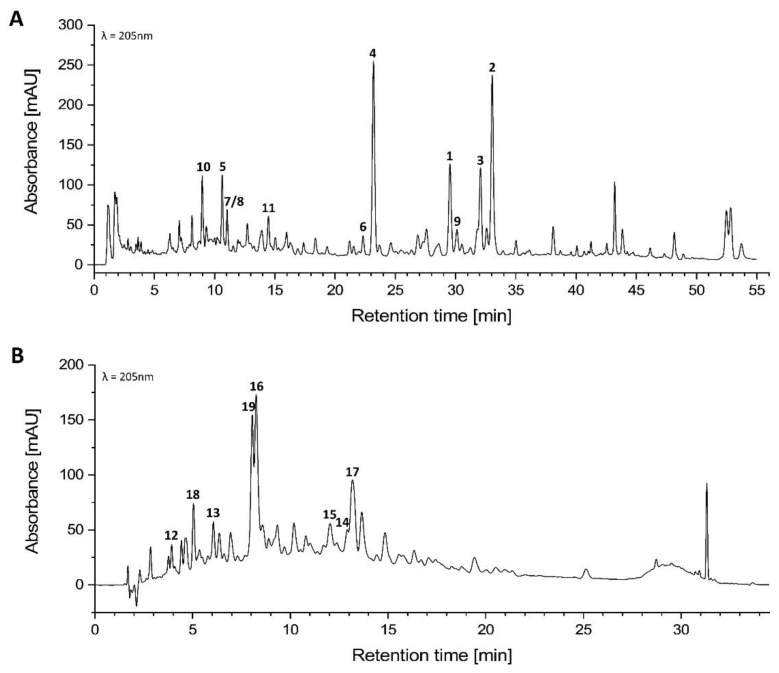
HPLC-UV chromatogram (205 nm) and peak assignment of the (**A**) PE fraction and (**B**) DEE fraction of the methanolic rhizomes and roots extract of *Cyperus articulatus*; for HPLC conditions, see Section 4.2 (General experimental methods). Due to a similar extract composition, the HPLC method for the DEE fraction of the methanolic rhizomes and roots extract, after DCM extraction of *Scirpoides holoschoenus*, was used for the DEE fraction of *C. articulatus* in (**B**) [47].

**Figure 3 molecules-27-05937-f003:**
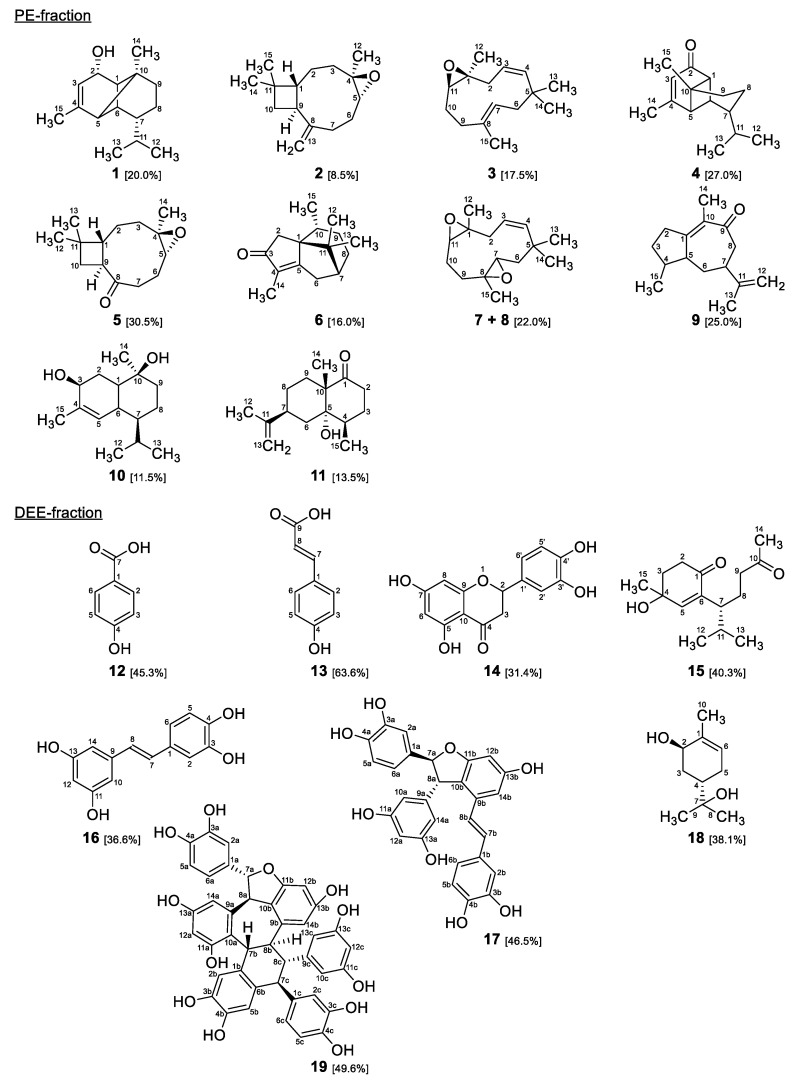
Compounds **1**–**19,** isolated from the subaerial parts from *Cyperus articulatus*. In brackets: NO production inhibition (% inhibition vs. LPS) of isolated compounds from J774A.1 murine macrophages at a test concentration 10 µM. Sesquiterpenes **1**, **3**–**5**, **9**, and **10**, phenolic acid compound **13,** and terpenoid compound **18** showed high antiproliferative properties in the performed MTT assay (Appendix A).

**Table 1 molecules-27-05937-t001:** NO production inhibition and iNOS and COX-2 expression inhibition vs. LPS ± SEM (*n* = 3) for 18 subfractions from the DEE fraction of the methanolic extract of *C. articulatus* at 5, 1, and 0.1 µg mL^−1^. Fractions indicated in bold were further investigated.

	NO Production(% Inhibition ± SEM vs. LPS)			iNOS Expression(% Inhibition ± SEM) vs. LPS)			COX-2 Expression(% Inhibition ± SEM) vs. LPS)		
DEE Fraction	5 µg mL^−1^	1 µg mL^−1^	0.1 µg mL^−1^	5 µg mL^−1^	1 µg mL-^1^	0.1 µg mL^−1^	5 µg mL^−1^	1 µg mL^−1^	0.1 µg mL^−1^
**A**	25.50 ± 0.37	14.83 ± 0.51	11.50 ± 0.16	39.90 ± 0.69	23.00 ± 0.35	19.00 ± 8.33	38.50 ± 9.50	23.50 ± 0.50	8.75 ± 0.50
**B**	30.67 ± 0.26	15.67 ± 0.91	12.00 ± 0.03	30.00 ± 0.88	11.33 ± 0.33	8.33 ± 0.33	34.50 ± 0.50	25.50 ± 0.58	23.75 ± 0.50
**C**	28.33 ± 5.29	18.33 ± 2.03	12.00 ± 2.52	45.50 ± 0.27	39.00 ± 0.00	33.67 ± 0.67	34.50 ± 0.42	27.50 ± 0.50	5.00 ± 0.45
**D**	27.55 ± 1.78	19.00 ± 1.50	15.89 ± 2.51	23.50 ± 0.42	12.67 ± 0.67	5.33 ± 0.33	28.00 ± 0.58	15.33 ± 0.33	10.00 ± 0.20
**E**	31.33 ± 2.52	22.67 ± 0.76	11.67 ± 0.88	35.50 ± 0.33	33.67 ± 0.67	29.67 ± 0.76	43.00 ± 0.52	24.00 ± 0.10	14.00 ± 0.50
**F**	29.67 ± 0.67	8.00 ± 0.17	2.00 ± 0.46	28.33 ± 0.33	22.00 ± 0.67	14.00 ± 0.00	26.50 ± 0.50	20.50 ± 0.10	13.25 ± 0.50
**G**	29.67 ± 0.43	22.00 ± 8.33	13.00 ± 6.51	37.67 ± 0.67	26.11 ± 6.08	19.67 ± 3.93	37.59 ± 0.18	17.17 ± 0.50	13.00 ± 0.30
**H**	32.33 ± 3.40	26.00 ± 0.08	21.67 ± 0.81	29.00 ± 0.00	27.00 ± 0.50	16.00 ± 0.50	27.75 ± 0.64	21.92 ± 0.42	10.92 ± 0.58
**I**	35.67 ± 0.18	14.33 ± 0.76	10.33 ± 0.76	31.50 ± 0.50	25.33 ± 0.33	16.00 ± 0.60	23.42 ± 0.42	12.33 ± 0.33	7.00 ± 0.00
**J**	23.00 ± 0.23	12.33 ± 0.67	10.00 ± 0.20	21.50 ± 0.50	11.00 ± 0.50	6.00 ± 0.33	34.75 ± 0.50	22.00 ± 0.30	15.00 ± 0.30
**K**	33.50 ± 0.50	14.50 ± 2.50	5.50 ± 0.50	33.00 ± 0.73	23.00 ± 0.10	14.00 ± 0.67	37.00 ± 0.50	28.43 ± 0.20	20.33 ± 0.33
**L**	27.00 ± 0.73	14.00 ± 0.19	5.33 ± 0.33	36.50 ± 0.50	24.50 ± 0.50	13.33 ± 0.33	24.25 ± 0.50	18.75 ± 0.50	9.33 ± 0.38
**M**	14.33 ± 0.84	3.33 ± 0.67	2.33 ± 0.38	35.00 ± 0.04	29.00 ± 0.67	19.00 ± 0.40	28.33 ± 0.33	23.00 ± 0.67	19.50 ± 0.50
**N**	57.67 ± 0.88	42.33 ± 0.64	36.00 ± 0.33	32.50 ± 0.50	27.00 ± 0.50	19.00 ± 0.40	33.67 ± 0.67	28.33 ± 0.33	17.00 ± 0.00
**O**	43.67 ± 0.55	39.33 ± 0.64	13.50 ± 0.89	35.50 ± 0.50	17.67 ± 0.33	13.50 ± 0.20	26.77 ± 0.71	19.33 ± 0.33	10.50 ± 0.20
**P**	22.33 ± 0.18	14.33 ± 0.08	6.00 ± 0.45	17.33 ± 0.50	14.00 ± 0.50	6.00 ± 0.50	40.33 ± 0.29	33.44 ± 0.55	24.00 ± 0.40
**Q**	36.67 ± 0.88	26.22 ± 0.20	25.33 ± 0.10	21.00 ± 0.58	13.33 ± 0.33	4.67 ± 0.67	27.67 ± 0.69	21.50 ± 0.42	4.50 ± 0.29
**R**	44.53 ± 0.53	27.67 ± 0.73	22.77 ± 0.33	34.33 ± 0.33	26.33 ± 0.20	24.67 ± 0.67	45.33 ± 0.42	27.33 ± 0.42	15.50 ± 0.42
**positive** **control**	**1 µM**			**1 µM**			**1 µM**		
L-NAME	44.50 ± 0.50			44.39 ± 0.44					
Indometacin							44.33 ± 0.33		

**Table 2 molecules-27-05937-t002:** NO production inhibition and iNOS and COX-2 expression inhibition vs. LPS ± SEM (*n* = 3) of compounds **12**–**19** from the DEE fraction of the methanolic extract of *C. articulatus* at 10, 5, and 1 µM.

	NO Production (% Inhibition ± SEM vs. LPS)			iNOS Expression (% Inhibition ± SEM) vs. LPS)			COX-2 Expression (% Inhibition ± SEM) vs. LPS)		
DEE Compound	10 µM	5 µM	1 µM	10 µM	5 µM	1 µM	10 µM	5 µM	1 µM
**12 (from E)**	45.26 ± 0.53	31.14 ± 0.55	10.34 ± 0.42	26.33 ± 0.10	17.46 ± 0.33	11.29 ± 0.32	28.14 ± 0.14	15.38 ± 0.52	4.00 ± 0.16
**13 (from E)**	63.64 ± 0.12	48.15 ± 0.15	29.36 ± 0.28	37.10 ± 0.67	30.65 ± 0.16	20.97 ± 0.25	33.97 ± 0.18	15.38 ± 0.62	10.27 ± 0.23
**14 (from E)**	31.41 ± 0.89	13.17 ± 0.72	5.42 ± 0.29	40.32 ± 0.58	32.25 ± 0.07	20.95 ± 0.25	15.38 ± 0.62	5.13 ± 0.05	2.27 ± 0.27
**15 (from F)**	40.27 ± 0.78	27.30 ± 0.67	18.94 ± 0.39	35.48 ± 0.18	24.19 ± 0.55	17.44 ± 0.40	26.92 ± 0.08	15.38 ± 0.46	8.06 ± 0.82
**16 (from H)**	36.58 ± 0.59	28.38 ± 0.31	19.42 ± 0.97	33.33 ± 0.33	23.98 ± 0.19	16.36 ± 0.04	32.00 ± 0.75	15.12 ± 0.20	12.21 ± 0.37
**17 (from M)**	46.50 ± 0.57	35.85 ± 0.24	31.65 ± 0.33	24.47 ± 0.42	14.65 ± 0.30	5.43 ± 0.33	36.03 ± 0.03	20.97 ± 0.19	14.21 ± 0.37
**18 (from N)**	38.10 ± 0.27	28.16 ± 0.57	19.70 ± 0.10	32.25 ± 0.07	14.96 ± 0.74	11.29 ± 0.32	15.38 ± 0.46	5.38 ± 0.39	0.00 ± 0.00
**19 (from P)**	49.63 ± 0.33	38.24 ± 0.10	30.33 ± 0.33	29.82 ± 0.56	17.54 ± 0.39	11.70 ± 0.49	27.91 ± 0.14	26.16 ± 0.94	10.78 ± 0.29
**positive** **control**	**1 µM**			**1 µM**			**1 µM**		
L-NAME	43.51 ± 0.33			45.23 ± 0.53					
Indometacin							40.60 ± 0.14		

## Data Availability

The data are contained within the article.

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
