# Peer review of "Bioassay-Guided Isolation of Anti-Inflammatory Constituents of the Subaerial Parts of Cyperus articulatus (Cyperaceae)"

_molecules, 2022, doi:10.3390/molecules27185937_

Round 1

Reviewer 1 Report

Mittas et al identified several bioactive molecules from C. articulants methanolic extracts of rhizomes and roots and demonstrated their potential anti-inflammatory activity as inhibitors of NO, iNOS, and COX2 expression. The isolation and characterization of the purified entities are thoroughly done guided by bioactivity proving the use of such material in the region where it is native. In general, the article is well written. However, it was quite difficult to follow the discussion section. I will encourage the authors if they can mention the best activity concentration below the structure of the identified molecules. I understand there are three different bioactivity testing they have done, and stating the best for each or making a scoring metric and ranking the identified molecules would be a very good visual to get the gist of the  structural and activity landscapes. Some minor comments:

1.     Section 4.4.1: C. articulates should be italicized

2.     Can the authors discuss the inclusion of SEC in section 4.4.1 for fraction 2?

Reviewer 2 Report

Bioassay-Guided Isolation of Anti-Inflammatory Constituents from the Subaerial Parts of Cyperus Articulatus

Line 3 - put the family name in parentheses after the species name

Line 17 – in vitro – put in italic

Line 21 – (COX-2) – term in parentheses

Line 27 - In the keywords, remove the species name and put the popular name and also replace the word anti-inflammatory by another that is not in the title, but is part of the context of the article

Line 48 – μg/mL – change by μg mL-1

Line 60 – (TNF-) and (IL-6) in parenthesis

Have other compounds of the species been isolated and identified, in addition to those characterized by GC-MS? If yes, mention them in the introduction.

Line 78 – idem L. 48

L.82 – ....of C. articulatus, change for “from C. articulatus”

The visualization of figure 1 is bad. Improve image quality

L. 117 - It was not clear how the compounds were identified. Whether spectral analysis and/or comparison with literature data was used. Make this clear in the text.

L.122 - And the analysis by GC. What parameter was used to identify the compounds?

L.138 - The parentheses after A and B got confused

L.144 – rewrite “of the subaerial parts from .....”

Correct the concentration units in table 1

Why the concentration units in table 1 are in micrograms per mL and in table 2 in micromols?

Is there no way to standardize everything to micromols?

L. 156 – in vitro  - put in italic

L. 223 - from the rhizomes and roots of C. articulatus – put “of the rhizomes and roots from C. articulatus”

L.255 – DAD detector

L. 262 e 263 – put mg mL-1 and other

Line 295 - Were the roots and rhizomes extracted together?

                    Was the UV at a fixed wavelength?

L. 430 – in vivo  - put in italic

L. 432 - The anti-inflammatory activity found is really interesting, but it still doesn't give enough support to corroborate the use of the species. Other tests would be necessary, such as the in vivo toxicity of the molecules, the development of pharmaceutical forms to verify the permanence of the activities. Redo this part.

Put all spectra in the supplementary material

Reviewer 3 Report

Dear Editor,

Molecules

I reviewed the article "Bioassay-Guided Isolation of Anti-Inflammatory Constituents from the Subaerial Parts of Cyperus Articulatus" this article demonstrate that phytochemical in- 16 vestigations of the methanolic rhizomes and roots extract of Cyperus articulatus, monitored by in 17 vitro assays, afforded 12 sesquiterpenes, three stilbenes, two phenolic acids, one monoterpene and 18 one flavonoid. Among them, 14 compounds were isolated for the first time from this species. Their 19 inhibitory potential against the nitric oxide (NO) production, as well as the inducible nitric oxide 20 synthase (iNOS) and the cyclooxygenase (COX)-2 expression, was evaluated in LPS-treated J774A.1 21 murine macrophages. Especially, stilbene dimer trans-scirpusin B and trimer cyperusphenol B 22 showed promising inhibitory activities on the production of the inflammatory mediator NO in a 23 concentration-dependent manner (10–1 µM). The obtained data are the first results confirming the 24 anti-inflammatory potential of C. articulatus and support its indigenous use as a traditional remedy 25 against inflammation related disorders

The manuscript includes a good scientific effort and I recommended it for publication in Molecules with minor revisions:

-1H, 13C-NMR and Mass spectrum of the compounds isolated should be included in supplementary data.
